# Bone Health in Mood Disorders: A Narrative Review about Clinical and Biological Connections

Antonella Maria Pia De Novellis [1,2,*], Giulia Ferrazzi [3], Gian Maria Galeazzi [1,4], Mattia Marchi [3,4], Matteo Meloni [1,2], Luca Pingani [1,4] and Silvia Ferrari [1,4]

1   Department of Biomedical, Metabolic and Neural Sciences, Section of Clinical Neurosciences, University of Modena & Reggio Emilia, 41125 Modena, Italy; gianmaria.galeazzi@unimore.it (G.M.G.); matte_meloni@yahoo.it (M.M.); luca.pingani@unimore.it (L.P.); silvia.ferrari@unimore.it (S.F.)
2   Department of Mental Health, Local Health Agency of Modena, 41124 Modena, Italy
3   PhD School in Neuroscience, Department of Biomedical, Metabolic and Neural Sciences, University of Modena & Reggio Emilia, 41125 Modena, Italy; giulia.ferrazzi@unimore.it (G.F.); mattia.marchi@unimore.it (M.M.)
4   Department of Mental Health, Local Health Agency of Reggio Emilia, 42122 Reggio Emilia, Italy
*   Correspondence: amp.denovellis@gmail.com

**Abstract:** Evidence about bone health in people affected by psychiatric disorders is limited. This narrative review aims to highlight what is known, up to the present time, about clinical connections between bone health and psychiatric disorders, particularly depressive disorders (DD) and bipolar disorders (BD), in terms of common biological pathways. Besides inflammation, we focused on two molecules of growing interest: neuropeptide Y (NPY) and the neuro-hormone melatonin. Also, the role of psychoactive drugs on bone tissue was explored. For the preparation of this narrative review, the scientific literature of the most recent 7 years from PubMed, Springer Nature, Science Direct (Elsevier), Wiley Online, ResearchGate, and Google Scholar databases was analyzed. Reviewed evidence reveals that people diagnosed with BD or DD have an increased risk of both fractures and osteoporosis; NPY reduces bone loss induced by longer periods of depression and "buffers" psychological stress effects on bone health. MLT shows beneficial effects in osteoporosis and bone healing. Lithium, a mood stabilizer, shows potential bone-protective activity, while antipsychotic and antidepressant treatments may increase the risk of bone tissue damage, though further investigation is needed.

**Keywords:** bipolar disorder; depressive disorder; bone healing; osteoporosis; neuropeptide Y; melatonin; lithium

## 1. Introduction and Background

### 1.1. Mood Disorders

Mood disorders are described as clinically relevant disruptions in emotions. These include bipolar disorder, cyclothymia, hypomania, major depressive disorder, disruptive mood dysregulation disorder, persistent depressive disorder, and premenstrual dysphoric disorder [1].

More specifically, bipolar disorder (BD) is a long-term condition characterized by mood fluctuations with clinical relevance. Its manifestation consists of recurrent episodes of mania or hypomania alternating with depressive episodes. The lifetime prevalence of bipolar spectrum disorders is about 2.4% [2]. BD is associated with a complex diagnosis, often misdiagnosed initially. Its management and prognosis are also complex, and it is associated with a higher risk of early mortality than the general population [3]. BD is associated with medical comorbidities, such as cardiovascular, metabolic, neurological, and other diseases [4].

A recent field of investigation is comorbidity with poor bone health. Only little is known about the relationship between BD and bone health: "external" factors such as lifestyle factors (smoking, alcohol, diet, physical activity, and socioeconomic status), prescribed drugs, and biomarkers of systemic inflammatory and oxidative stress processes have a widely demonstrated influence on both mood disorders and bone diseases. However, evidence about what affects the connection between these conditions is still lacking [5].

Depressive disorder (DD) is characterized by recurrent episodes of a depressed mood, loss of pleasure and interest in daily activities, loss of motivation, lack of physical energy, neurovegetative symptoms, such as sleep disturbance and appetite alteration, feelings of hopelessness, and lack of concentration [6]. It represents one of the primary causes of disability in the world [7]. Its prevalence is estimated at about 5% of the global population, resulting from a complex interaction of social, psychological, and biological events [8]. There is large evidence about the associations between DD and various somatic diseases, such as cardiovascular diseases [9], metabolic diseases [10], neurological diseases [11], cancer [12], immune-mediated inflammatory diseases [13], and chronic respiratory diseases [14,15].

Major DD was also associated with a clinically significant lifetime increased risk of fractures, suggesting that DD should be considered among the risk factors for bone lesions, together with others [16].

### 1.2. Bone Healing and Osteoporosis

Bone healing is the complex regenerative process of bone tissue after its fracture. The healing process via intramembranous ossification has little or no inflammatory response, and it is very slow. It takes from a few months to a few years to achieve complete regeneration. This process is called primary or direct bone healing, and it is less common than the secondary or indirect one. Primary bone healing consists of the direct differentiation of the mesenchymal stem cells (MSCs) into osteoblasts, which in turn deposit mineralized extracellular matrices [17]. Secondary bone healing is the most common form of fracture regeneration, and it is faster than the primary one; it is characterized by three overlapping phases: the inflammatory phase, the repair phase, and the remodeling phase [18]. Immune cells and MSCs participate in critical cellular pathways and communications to control the bone healing process [19].

Osteoporosis is a skeletal disorder characterized by reduced bone strength, and it represents a risk factor for bone fractures [20]; it should be conceived as a systemic metabolic disease [21]. The osteoporotic bone shows significant differences in microstructure compared to normal controls, with reduced bone mass and destruction of the bone microstructure [22]. Moreover, bone remodeling differs in osteoporotic people compared to healthy subjects: affected individuals may show low, normal, or increased bone remodeling, depending on the etiology of osteoporosis [23].

Bone mineral density (BMD) is a measure of bone mineral content directly related to bone strength [24]. BMD at either the spine or hip bone predicts the overall risk of fractures; the lower the BMD, the higher the relative fracture risk.

In recent years, the investigation of the pathogenesis process of osteoporosis has become an interesting field of research [25]. Osteoporosis is a complex and multi-factorial illness related not only to hormonal alterations but also to the interplay of other risk factors, such as genetics, endogenous, or lifestyle factors [26]. Moreover, the study of the so-called "brain–gut–bone" axis revealed that brain pathways related to the gut microbiome may affect the occurrence and progression of osteoporosis [27]. The review by Zhang and colleagues discussed in detail and summarized the interplay factors involved in osteoporosis based on brain–bone, gut–bone, and brain–gut connections [28].

### 2. Materials and Methods

This narrative review aims to highlight the biological and clinical existing relationships between mood disorders, specifically BD and DD, and bone tissue health. The scientific

literature was analyzed over the last 7 years, searching PubMed, Springer Nature, Science Direct—Elsevier, Wiley Online ResearchGate, and Google Scholar databases, using a combination of terms related to ("bone healing" OR "bone fractures" OR "bone tissue health") AND ("osteoporosis" OR "bone mineral density") AND ("mood disorders" OR "bipolar disorder" OR "depressive disorder") AND ("NPY" OR "Neuropeptide Y" OR "Melatonin"). A report on recent evidence about bone-related comorbidities in people diagnosed with BD and DD is provided (Section 3.1), including evidence supporting the increased risk of fractures and osteoporosis and the interplay roles of neuropeptide Y (NPY) and melatonin (MLT) in mood disorders. In Section 3.5, the potential effects on bone tissue health of psychoactive drugs prescribed for mood disorders are reported, with specific sub-paragraphs for lithium, antipsychotics (APs), and antidepressants (ADs), respectively.

## 3. Results

### 3.1. How Are Mood Disorders and Bone Tissue Health Related?

3.1.1. Increased Risk of Fractures and Osteoporosis in Patients with BD and DD

People suffering from DD reported higher risks of falls and fractures compared with the general population, as shown by Shi and colleagues, who found a significant association between DD and an increased risk of hip fractures, which was stronger in European and male patients with DD [29]. A longitudinal study investigated the relationship between baseline depression and subsequent accidental and/or unexplained falls at a 2-year follow-up in people older than 50 years. The association was demonstrated, as well as the increased risk of future falls. This evidence is stronger for a link between DD and unexplained falls [30]. A previous meta-analysis of 14 prospective studies already confirmed that depressive symptoms increased the risk of falls by almost 50% in older people; it also found that DD is a strong predictor of falls in nursing home residents [31]. A more recent systematic review and meta-analysis revealed that the prevalence of DD among older patients with hip fractures was 23% [32]. Also, the meta-analysis of prospective studies conducted by Wu and colleagues in 2018 confirmed the significant association between DD and the increased risk of fracture and bone loss, underling the need for prevention strategies [33]. The risk of hip fracture was also found to increase in patients with major depressive disorder compared to healthy individuals in the national Korean cohort study conducted by Kim and colleagues [34]. Another recent study confirmed the evidence of the association between DD and osteoporosis and estimated that DD should be considered a risk factor for osteoporosis, such as other known predictive conditions [35]. Also, the risk of osteoporosis is a consequence of lifestyle factors, such as limited physical activity, smoking, or diet (i.e., low calcium and phosphorus intake, alcohol consumption, and altered intake of proteins) [36]. These same factors are also known for their complex bidirectional association with DD; therefore, it is also through lifestyle-related aspects that the link between osteoporosis and mood is explained [37–40].

Focusing on BD, a few previous studies showed an increased risk of fractures and lower BMD in this group of patients [41]. Most recently, the systematic review by Chandrasekaran and colleagues remarked that people with BD, independent of age, sex, comorbidities, and medication use, had an increased risk of fracture [42]. These findings seemed related to common pathways that promote bone loss and cause neuro-progression in BD, such as inflammation, mitochondrial dysfunction, oxidative stress, and endocrine factors [3]. The large cohort of Stubbs and colleagues' study confirmed the increased risk of hospitalized falls and hip fractures in older adults with BD and other mental illnesses [43]. In the cohort study of Ma and colleagues, the predictors of falls and fractures causing hospitalization in people diagnosed with mood disorders over a 5-year follow-up were investigated: approximately 8% of patients affected by mood disorders were hospitalized with a fall or fracture; similar factors, such as older age, comorbid physical disorders, and analgesic use, were found to predict the risk of falls and fracture [44]. A study by Li and colleagues aimed to assess bone mineral density and related influencing factors in a sample of people recently diagnosed with BD and without any drug treatment; the results were age- and sex-matched

with healthy controls [45]: drug-free patients affected by BD had significantly lower BMD in comparison to healthy controls in multiple body regions. The recent case-control study by Williams and colleagues confirmed the association between BD and poor bone quality and quantity [5]. Another recent longitudinal register-based study was conducted in Denmark by Köhler-Forsberg and colleagues. It revealed that BD is associated with an increase in the risk of osteoporosis. It also showed that lithium treatment decreases the risk of osteoporosis in this subgroup of patients, so lithium may have a protective role [46].

3.1.2. Molecules Involved in Mood Disorder Genesis and/or Clinical Presentation with Effects on Bone Tissue Health

The possible mechanism underlying the increased risk of fracture in patients with BD or DD involves the interplay of many factors. One of them is the interaction between inflammatory cytokines and osteoclasts, such as the tumor necrosis factor-alpha (TNF-$\alpha$), which is a crucial cytokine that not only appears to be involved in the pathogenesis of BD but also influences bone resorption [47,48]. Despite the controversy of the evidence, the pathogenesis of BD seems related to immune dysregulation in both the central and peripheral nervous systems [49]. The bidirectional regulation of the bone–brain axis consists of different regulatory effects of bone-derived cytokines and bone-derived cells, not only on individual brain function but also on the occurrence of related psychiatric disorders [50]. In the wide field of research on the inflammatory hypothesis of mood disorder, several clinical trials attested to the antidepressant efficacy of anti-TNF-$\alpha$ molecules in patients with medical illnesses and diagnosed with DD or bipolar depression. The results showed that the peripheral inhibition of TNF-$\alpha$ activity may decrease brain inflammation; more evidence about the effectiveness of selective TNF-$\alpha$ antagonists as a treatment for mood disorders is needed [51].

The oxidative stress process may provide a further biological link between both bone functioning and mood disorders. After fractures, especially in the early phase, inflammatory and ischemic conditions lead to the secretion of reactive oxygen species, but sometimes intrinsic oxidative stress can cause excessive toxic radicals, resulting in irreversible damage to cells responsible for bone healing [52]. Oxidative stress is also known to be involved in stress-related mood disorders: the review by Kalinichenko and colleagues attested to changes in the pro-oxidant/antioxidant levels in patients with DD [53].

Inflammatory processes are indeed very complex, with multiple biochemical variables affecting them. Figure 1 attempts to provide a sense of this complexity by describing the most relevant inflammatory processes influencing both the brain and the bone.

Neuropeptide Y and melatonin appear to play a determinant role in these complex interactions. The following two sections provide evidence of their potential effects on mood disorders and bone tissue regulation.

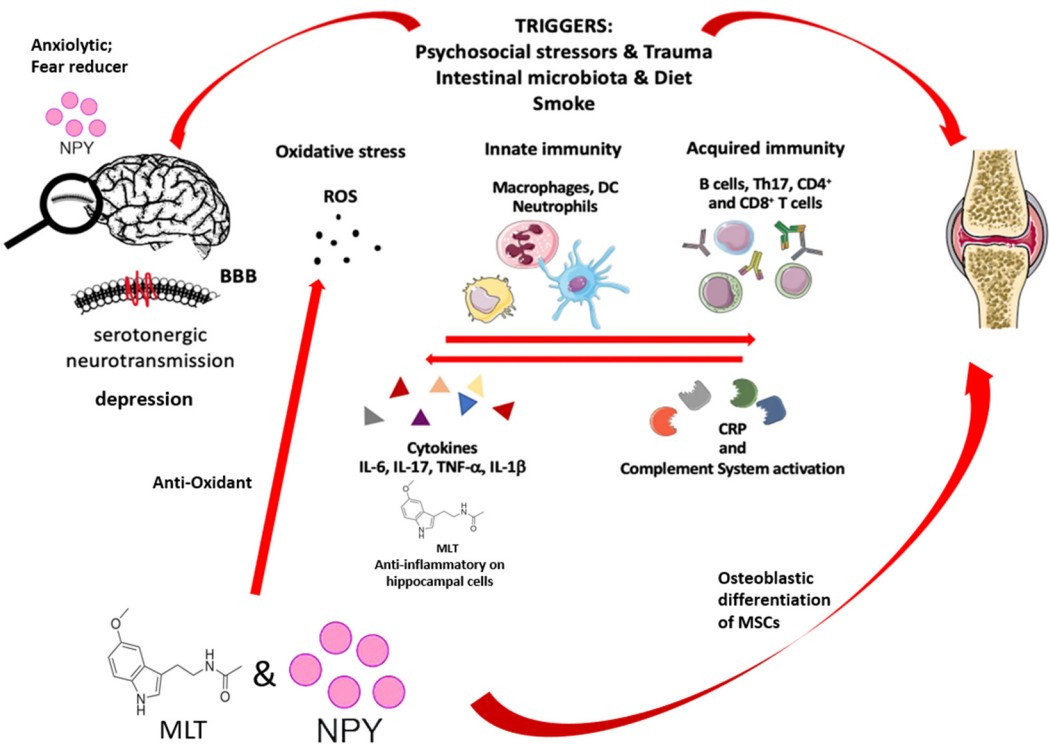

**Figure 1.** Inflammatory pathways involved in the development of mood symptomatology and bone disease, including NPY and melatonin role. (Abbreviations: BBB = blood–brain barrier, ROS = reactive oxygen species, DC = dendritic cells, IL = interleukin, TNF = tumor necrosis factor, CRP = C-reactive protein, MSCs = mesenchymal stem cells (Courtesy of Chimenti et al., 2021, modified [54])).

### 3.2. Neuropeptide Y (NPY)

Although the evidence about inflammatory processes in mood disorders and related signaling molecules is vast [55], less is known about NPY and its receptors.

NPY is a neurotransmitter acid peptide that is widely distributed both in many brain regions and in the peripheral sympathetic nervous system. The highest central concentration includes the paraventricular hypothalamic nucleus, hypothalamic arcuate nucleus, suprachiasmatic nucleus, median eminence, dorsomedial hypothalamic nucleus, paraventricular thalamic nuclei, amygdala, hippocampus, the nucleus of the solitary tract, locus coeruleus, nucleus accumbens, and cerebral cortex. In the peripheral sympathetic nervous system, NPY is located and released together with noradrenalin, and the adrenal medulla is, therefore, the major storage site of NPY [56]. NPY is also produced in many other organs, such as the liver, heart, spleen, kidneys, urogenital tract, and endothelial cells of blood vessels [57]. Furthermore, NPY acts as a gastrointestinal hormone.

NPY has been studied in DD as a diagnostic biomarker [58] and in BD as a suicide-attempt marker [59]. NPY seems to be primarily affected in DD, and it contributes to its clinical presentation [60]. NPY is part of the central melanocortin system, a circuitry that also involves other peptides, such as pro-opiomelanocortins and agouti-related proteins. This system has shown effects on anxiety and depressive-like behaviors [61]. An interesting study on animal models showed that ADs, despite many other effects, increase NPY expression levels [62], so it can also be considered a treatment-response biomarker. Moreover, a recent double-blind, placebo-controlled trial was conducted to investigate the antidepressant effect of a single dose of intranasal NPY itself. It showed that NPY decreases the symptomatology in subjects affected by major depressive disorder [63]. Furthermore, NPY not only demonstrated a stress-protective role through its Y2 receptors [64] but also seemed to be a neuromodulator with anxiolytic and fear-reducing effects [65]. Its function has also been related to cognition and neurogenesis [66]. Its role in affecting the microbiome–gut–brain (MGB) axis is under further investigation in cognitive pro-

cesses related to schizophrenia and severe mood disorders, more specifically BD and major DD [67,68]. A systematic review and meta-analysis found that gut microbiota perturbations were associated with a transdiagnostic pattern, a depletion of certain anti-inflammatory bacteria, and an enrichment of pro-inflammatory bacteria in patients with DD and BD [69]. The recent study conducted by Sharma and colleagues, which analyzed postmortem brains of adult healthy controls and depressed suicide subjects, also found alterations in the expression of NPY and its four receptors (named NPY1R, NPY2R, NPY4R, and NPY5R) in suicides, as compared with healthy subjects. More specifically, a significant decrease in the NPY protein and mRNA expression in two different brain regions, the prefrontal cortex and hippocampus, of depressed suicide people was reported [70].

NPY has many metabolic roles on the bone tissue as well, both directly and indirectly. NPY directly regulates bone formation and resorption; indirectly, it participates in bone metabolism, affecting gut microbiota and blood vessel formation, and plays an intermediary role in autonomic nerve regulation on bone remodeling processes [71]. Of interest, a correlation analysis by Xie and colleagues showed that the changes in gut microbiota were closely associated with bone microstructure and serum Ca2+ levels. This study also revealed that Y1R antagonists may play an anti-osteoporotic effect in animal models [72]. Moreover, it has been demonstrated that NPY could accelerate the osteoblastic differentiation of mesenchymal stem cells, and it has been shown that NPY levels increase during early fracture healing, promoting this process. The study by Gu and colleagues provided clinical evidence of NPY's contribution to post-fracture bone healing in patients with traumatic brain and fracture combined injuries. It also suggested that NPY may represent a prognosis predictor and a potential therapeutic intervention in individuals with fractures [73]. Other animal studies confirmed NPY's role in fracture healing [74]. Moreover, NPY showed a regulative effect on bone marrow MSCs, thus influencing the development of osteoporosis and osteoarthritis [33].

The chronic psychological stress associated with depression may also lead to substantial bone loss. NPY pathways have a critical role, acting in the hypothalamus to attenuate this bone loss induced by longer periods of depression. NPY, acting via Y2 receptors in both central and peripheral neural tissues, inhibits noradrenaline and corticosterone production and release to protect bone from the negative effects of stress. These insights suggest a potential opportunity for therapeutic design that accounts for both primary mood disorders and general side effects, potentially reducing negative effects on bone health [75]. Moreover, the review of Ng and Chin investigated the relationship between psychological stress and bone health and revealed that the NPY and other molecules involved have a "buffer" role during mental stress, affecting the bone tissue [76].

### 3.3. Melatonin (MLT)

MLT is considered a neurohormone, mainly produced by the human pineal gland. Small amounts of MLT are also produced by other sites, such as the retina, lymphocytes, bone marrow, the gastrointestinal tract, and the thymus. MLT has a discontinuous secretion, responding to the circadian clock stimulation, regulated by the suprachiasmatic nucleus (SCN) in the pineal gland in dark conditions. It also has a continuous secretion via the cutaneous melatoninergic system [77]. Melatonin can reach different body sites from the blood and acts thanks to its receptors, the most important of which are MLT-receptor 1 (MT1) and MLT-receptor 2 (MT2). Thanks to MT1 and MT2, MLT can regulate sleep patterns: its onset is the starting point of the biological night [78]. MLT also sets circadian rhythms of many biological processes; thus, secondary feedback generates cross-talks between the circadian clock and metabolism, involving the bioactivity of MLT in antioxidant processes and immune regulation. MLT shows an important anti-neoplastic action [79], and it may influence reproductive processes and play an important role in mood disorders [80,81].

Decreased MLT levels have been found in patients affected by DD, and a link between low brain MLT and serotonin levels in the brains of patients suffering from DD has been reported [82]. The disruption of the melatonin system might be one of the core mechanisms

underlying the pathophysiology of depressive disorder, given melatonin's role as an anti-inflammatory and neuroprotective neuro-hormone of hippocampal cells [83]. Of interest, Dmitrzak-Weglarz and colleagues investigated the melatonin pathway in healthy control subjects and in patients affected by unipolar and bipolar depression: they demonstrated an existing dysfunction in the molecular regulation of the melatonin biosynthesis pathway; moreover, they revealed a specific pattern for unipolar and bipolar depression at the genetic level, with their polymorphisms, and at serum biomarkers' examination [84].

In patients with BD, a significantly decreased MLT secretion is revealed, especially in the depressed phase of the disease. On the contrary, melatonin secretion returns to normal levels when the symptoms are remitted, and patients with BD showed abnormal oscillating melatonin secretion [81,85]. Another piece of evidence that decreased the production of melatonin, interacting with other factors, seems to be involved in the pathogenesis of BD is the study by Carta and colleagues [86]. Finally, there is much evidence of abnormalities in the biological clock functioning and melatonin levels in individuals affected by mood disorders, with differences between patients with bipolar and depressive disorder [87,88].

In the past century, due to its multiple neurobiological actions, MLT has gained growing interest as a potential treatment not only for various inflammatory diseases [89] but also for mood disorders [90]. This was only partially confirmed: melatonin seems to be a promising intervention in the adjunctive treatment of bipolar mania, and its role in symptoms of sleep disturbance, mania, and depression in people diagnosed with BD was confirmed [91].

MLT also has important effects on bone tissue: In animals, MLT plays a role in promoting the osteoblastic differentiation of MSCs and in fracture healing through NPY and its Y1 receptor [92]. A recent review confirmed that MLT has beneficial effects in bone- and cartilage-related disorders, such as osteoporosis and bone fracture healing [93]. These findings can be relevant when the "vicious cycle" involving depression and osteoporosis is considered, suggesting that MLT use in DD and related sleep disturbance symptoms may ameliorate the global quality of life of depressed people, not only contributing to symptom control but also decreasing the risk for osteoporosis [94,95].

### 3.4. Mood Disorders and Bone Disease: The Role of Psychomotor Performance

Psychomotor performances may also play a significant role in explaining the impact of mood disorders on bone health. Both depressive and bipolar disorders, in fact, and especially during acute decompensation, cause significant worsening of individual functioning, including psychomotor performances, with compromised mobility and possible effects on bone health regulation.

Furthermore, a deficiency in vitamin D and high levels of cortisol and inflammatory cytokines are often documented in depressive episodes as well as osteoporosis and cognitive decline [96–98]. This evidence is the result of the interaction among different variables such as hormonal imbalance, physical activity, antidepressants, and hormonal medicaments. In a recent population-based study, Mehta and colleagues found that the BMD of the hip bone was associated with overall cognitive function, while mood disorders were not associated: this suggests that individuals with a lifetime history of mood disorder had a poorer cognitive function and increasing BMD and that it is necessary to better explore the association between cognitive function and possibly modifiable health conditions [99].

### 3.5. Psychoactive Drugs: Positive or Negative Effects on Bone Health?

Mood disorders such as DD and BD are chronic, multi-factorial, and disabling conditions with an uncertain etiology: their therapeutic management almost invariably implies pharmacological interventions, often with more than one drug and for a long period; pharmacological side effects can be multiple and complex and can appear in the short- and/or long-term of the treatment duration. Therefore, a focus on the possible impact of medications commonly used for DD and BD, such as lithium and ADs, on bone tissue health is of great significance.

### 3.5.1. Lithium Treatment

Lithium is the first-choice treatment for the prevention of manic and depressive episodes in BD. It is also the first-choice drug to treat manic episodes in the acute phase; finally, it can be used in augmentation to ADs for treatment-resistant depression [100–103]. In long-term treatment, lithium showed effectiveness and a relatively well-tolerated profile in older adults with BD and treatment-resistant major depressive disorder [104]. Lithium also shows immunomodulatory and antiviral properties and is the mood-stabilizing drug with the highest effectiveness in preventing suicidal behaviors [105].

The biochemical mechanism of lithium is associated mainly with the inhibition of glycogen synthase kinase-3 (GSK3β) and its effects on intracellular signaling, but its exact mechanism of action is still not fully understood [106]. GSK3β inhibition also leads to the activation of the WNT signaling pathway, a process also associated with bone formation [107,108]. Animal studies have shown that lithium enhances bone formation and fracture healing [109,110]. Recent studies in humans had associated lithium medication, particularly when administered for extended periods, with a higher BMD and a decreased risk of osteoporosis-related fractures [111,112]. The recent longitudinal register-based study conducted by Köhler-Forsberg and colleagues also revealed a cumulative dose-response relationship between lithium treatment and a decrease in the risk of osteoporosis. It is also reported that the longer the duration of lithium medication, the greater the reduction in osteoporosis risk, suggesting a potential bone-protective effect of lithium. Furthermore, this study demonstrated that other commonly used treatments for BD, such as the mood stabilizers valproate and lamotrigine, did not show the same protective effect on bone health as lithium [46]. However, it should be noted that a recent work by Hafizi and colleagues reported contrasting findings: lithium treatment was found to be associated with poorer bone texture, as revealed by measuring the lumbar spine trabecular bone score (TBS) in women diagnosed with mental disorders and with psychotropic medication intake [113]. Further investigations about the role of lithium on bone tissue are needed, based on the study of the texture of bone tissue, since the BMD only could potentially underestimate the fracture risk related to certain medicaments.

### 3.5.2. Antipsychotics (APs)

Despite controversial evidence, AP medications are often used in DD and BD, especially in severe or psychotic manifestations [114]. AP drugs are associated with decreased BMD and an increased risk of fractures and osteoporosis. This may happen through dopamine, serotonin, and adrenergic receptor signaling since these receptors were found on osteoclasts and osteoblasts [115]. The recent review conducted by Weerasinghe and colleagues provided an overview of first-, second-, and third-generation antipsychotics' effects on bone formation and resorption, influencing the expression profiles of dopamine, serotonin, and adrenergic receptors through intracellular pathways [116]. The study by Raffin and colleagues highlighted that, when exposed to iatrogenic hyperprolactinemia caused by second-generation APs, young psychiatric patients may face a cumulative risk of osteoporosis, suggesting that the secretion and activity of prolactin, as well as its balance with vitamin D, are also involved in the complex interplay of factors connecting mood disorders and bone health and may become clinical indicators [117]. APs differ remarkably from one another in terms of effects on bone tissue: among second-generation APs, aripiprazole showed a more favorable profile [117,118]. A systematic re-evaluation of prolactin levels in patients undergoing AP treatment is recommended for the implications on bone health [118].

### 3.5.3. Antidepressants (ADs)

AD medications are the first-line treatment for DD, but they are also largely prescribed for bipolar depression. Among the ADs, serotonin selective reuptake inhibitors (SSRIs) represent the most commonly used [119]. Since osteoblasts, osteoclasts, and osteocytes are all provided with the serotonin transporter (5-HTT) and receptors (5HTR), SSRIs may

affect their functioning [120]. In an interesting review, the association between ADs (both SSRIs and tricyclic ADs) and lower BMD was shown to have an increased risk of fracture, especially within the first 14 days of use [121]. On the other hand, the meta-analysis conducted by Schweiger and colleagues showed no difference in BMD between AD users vs. non-users and remarked that evidence is very limited to draw solid conclusions [122], prompting the need for further research on this topic.

Table 1 sums up evidence on the effects of psychoactive medications on bone tissue.

**Table 1.** Effects of psychoactive drugs on bone tissue.

| Class of Psychoactive Drug | Psychoactive Drug | Effects on Bone Tissue |
| --- | --- | --- |
| Mood stabilizers | Lithium | • Enhances bone formation and fracture healing;<br>• Is associated with increased BMD;<br>• It is associated with a decreased risk of osteoporosis-related fractures. |
| | Valproate and Lamotrigine | • Did not show effects on bone tissue. |
| APs | First generation APs | Are associated with:<br>• Decreased BMD;<br>• Are associated with an increased risk of fractures and osteoporosis. |
| | Second generation APs | • Cause iatrogenic hyperprolactinemia leading to cumulative risk of osteoporosis;<br>• Aripiprazole showed a more favorable profile. |
| ADs | SSRI and Tricyclic ADs | Are associated with:<br>• Lower BMD;<br>• Increased risk of fracture, especially within the first 14 days of use;<br>• There is evidence of no differences between Ads users and not-users. |

(Abbreviations: BMD = bone mineral density; AP = antipsychotic; AD = antidepressant; SSRI = selective serotonin reuptake inhibitors).

## 4. Limitations of the Study

The main limitation to be acknowledged is the very limited scientific evidence already available on this topic, despite its promising clinical relevance: the review succeeded in including virtually the whole of the evidence that is so far available. Furthermore, the heterogeneity of the selected studies may have added to variability in reported estimated connections, suggesting again the need for a more systematic research approach to this topic. Finally, the study focused on mood disorders and bone tissue connections, mostly exploring cellular and signaling processes. There are several psychological, medical, and hormonal factors that need to be adequately considered and assessed to make generalized and appropriate considerations.

## 5. Conclusions

There is existing scientific evidence related to bone health in people affected by DD and BD, suggesting that these patients are at higher risk of bone fractures and osteoporosis.

Bone health and psychiatric conditions or symptoms seem to be connected by hormonal and intracellular-signaling pathways, but up to now, very little is known about these connections. External factors such as lifestyle habits, prescribed drugs, systemic inflammation, and oxidative stress processes have a well-demonstrated influence on both mood disorders and bone diseases. However, evidence about the potential role of these external factors in the link between mood disorders and bone affection is still very limited. The relationships between NPY and mood disorder and between NPY and bone mass maintenance are intriguing and need further investigation [65]. Also, MLT may have a

crucial role, and further evidence for its future use in the treatment of fracture healing is needed [92].

Psychoactive drugs, such as lithium, APs, and ADs, also have an influence on bone processes. While lithium may have potential bone-protective properties, other psychoactive drugs show a negative interaction with bone tissue health. Further research is needed to better understand and manage the risk of osteoporosis in psychiatric patients receiving AP treatments [117]. In terms of clinical practice implications, BMD monitoring, and osteoporosis screening should be conducted regularly in patients with mood disorders, especially if older and with established comorbid chronic illnesses [44].

**Author Contributions:** All authors contributed significantly to the present work. A.M.P.D.N. conceived and designed the manuscript, conducted the literature search, interpreted the data for the work, and drafted the manuscript. S.F., G.M.G. and L.P. revised the work critically for important intellectual content. M.M. (Mattia Marchi), M.M. (Matteo Meloni) and S.F. conceived and designed the manuscript, interpreted the data, and helped edit the written manuscript. G.F. supervised the literature search and the formal analysis. S.F. and G.M.G. conducted the final supervision. All authors have read and agreed to the published version of the manuscript.

**Funding:** No external funding was received for this research.

**Institutional Review Board Statement:** The study was conducted according to the guidelines of the Declaration of Helsinki. Ethical review and approval were not applicable for studies not involving humans or animals.

**Informed Consent Statement:** Not applicable.

**Data Availability Statement:** No new data were created or analyzed in this study. Data sharing is not applicable to this article.

**Conflicts of Interest:** The authors declare no conflicts of interest.

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
