# Peer review of "Bone Health in Mood Disorders: A Narrative Review about Clinical and Biological Connections"

_2673-5318, doi:10.3390/psychiatryint5010006_

Round 1
Reviewer 1 Report
Comments and Suggestions for Authors
Manuscript by the author Antonella Maria Pia De Novellis and group is a narrative review which discusses about the current knowledge about the bone health in depressive and bipolar disorders. The authors argue that the possible connection between bone health and mood disorders, due to the lack of substantial scientific evidence, warrants increased attention in future research. The potential combined advantages of NPY and melatonin's roles are especially intriguing.
Overall, this review manuscript covers an interesting topic. It is adequately presented with recent and relevant citations.
I have only a few remarks/suggestions.
1. Authors indicated that Li treatment might provide some beneficial effects, however, a study by Hafizi et al., Bone 2022 showed that Li treatment could also be related to lowering bone density similar to the SSRIs and TCAs.
2. While the authors concentrated on the biological links between bone health and mood disorders, it's important to note that both types of mood disorders significantly affect goal-directed behavior, cognitive processes, and emotions. Consequently, this leads to a decrease in psychomotor performance, which involves reduced motor responses and related control. Therefore, these aspects should be adequately considered and assessed when endeavoring to establish a clinical and biological connection.
3. In their current state, the concise summary and the abstract seem redundant and fail to fulfill their intended purpose. Streamlining these two sections, along with the conclusion, could enhance the manuscript's overall appeal.
4. The seemingly unconnected exploration of inflammatory molecules in section 3.1.2, NPY in 3.2, and Melatonin in 3.3 introduces a level of distraction, which becomes more pronounced when considering Figure 1.
5. Figure 1 can be redesigned to provide a more holistic summary of the presented narrative by including NPY/MLT.
Reviewer 2 Report
Comments and Suggestions for Authors
The authors focus on bone health in mood disorders: a narrative review about clinical and biological connections. The study is interesting and fills the void in the field. The paper is well-designed, and the manuscript is well-written. However, the work has shortcomings that should be filled in and some errors to be corrected.
- The "abstract" section should consist of a single paragraph. There is no need to add a "simple summary" section – see template
- The statement in lines 64-68 is not entirely true. While knowledge about the links between BD and bone tissue disorders is incomplete, its links with lifestyle factors systemic inflammatory and oxidative stress markers have already been widely described.
- Lines 101-103 do not confuse bone turnover markers levels with bone tissue remodeling.
- The statement: "did not show the same protective effect on bone health as lithium " in Table 1 is too elementary
-
- The figures should be placed directly after being mentioned in the text – line 213
- There is no author contributions in the "author contributions" section
- Bibliography format is inconsistent and should be adjusted to the publisher's requirements
Comments on the Quality of English LanguageMinor editing.
Reviewer 3 Report
Comments and Suggestions for Authors
The topic of the paper is very interesting and it will be of interest and relevance to specialists in phychiatry as well as bone biologists.
The paper is well-written and it provides information on the most important interactions between psychiatric/mood disorders and bone tissue.
However, I believe that the Conclusions section should be expanded. Moreover, the paper would benefit if additional figures/schemes related to relations between NPY, MLT, lithium, APs and ADs and bone are added.
Minor typos: Check the number of sections in Results section. It appears that 3.4. is missing.
Comments on the Quality of English Language-
